# Antimicrobial Resistant *E. coli* in Pork and Wild Boar Meat: A Risk to Consumers

**DOI:** 10.3390/foods11223662

**Published:** 2022-11-16

**Authors:** Martina Rega, Laura Andriani, Silvia Cavallo, Paolo Bonilauri, Silvia Bonardi, Mauro Conter, Ilaria Carmosino, Cristina Bacci

**Affiliations:** 1Food Hygiene and Inspection, Veterinary Science Department, University of Parma, Strada del Taglio, 10, 43126 Parma, Italy; 2Istituto Zooprofilattico Sperimentale della Lombardia ed Emilia Romagna, via Pitagora, 2, 42124 Reggio Emilia, Italy; 3Istituto Zooprofilattico Sperimentale della Lombardia ed Emilia Romagna, via Emilio Diena, 16, 41122 Modena, Italy

**Keywords:** AMR, quinolones, aminoglycosides, β-lactams, pork, wild boar, meat samples

## Abstract

Antimicrobial-resistant foodborne microorganisms may be transmitted from food producing animals to humans through the consumption of meat products. In this study, meat that was derived from farmed pigs and wild boars was analyzed and compared. *Escherichia coli* (*E. coli*) were isolated and tested phenotypically and genotypically for their resistance to quinolones, aminoglycosides and carbapenems. The co-presence of AMR-associated plasmid genes was also evaluated. A quinolone AMR phenotypic analysis showed 41.9% and 36.1% of resistant *E. coli* derived from pork and wild boars meat, respectively. A resistance to aminoglycosides was detected in the 6.6% of *E. coli* that was isolated from pork and in 1.8% of the wild boar meat isolates. No resistant profiles were detected for the carbapenems. The quinolone resistance genes were found in 58.3% of the phenotypically resistant pork *E. coli* and in 17.5% of the wild boar, thus showing low genotypic confirmation rates. The co-presence of the plasmid-related genes was observed only for the quinolones and aminoglycosides, but not for the carbapenems. Wild boar *E. coli* were the most capable to perform biofilm production when they were compared to pork *E. coli*. In conclusion, the contamination of pork and wild boar meat by AMR microorganisms could be a threat for consumers, especially if biofilm-producing strains colonize the surfaces and equipment that are used in the food industry.

## 1. Introduction

Antimicrobial drug use is the most important factor leading to antimicrobial resistance phenomenon, and Multi-Drug Resistance (MDR) is considered a significant hazard to public health [1,2]. This global public health threat causes 670,000 resistant infections and 33,000 deaths in Europe each year [3]. 

Quinolone resistance is related to the mutations and acquisition of resistance-conferring genes. The mutations are localized in one domain of the GyrA and ParE subunits of the respective enzyme, causing a reduction in the drug affinity for the enzyme–DNA complex [4,5,6]. Low-level resistance can be driven by plasmid-acquired resistance genes (Plasmid Mediated Quinolone Resistance-PMQR) that encode for a Qnr protein that protect the target enzyme from the quinolone’s activity [7]. Quinolone antimicrobial resistance can be conveyed by other plasmid resistance genes that can codify for a mutant aminoglycoside modifying enzyme (AME), thus conferring resistance to both of the antimicrobial classes simultaneously [6,8]. Several studies have evaluated the simultaneous presence of quinolone resistance and aminoglycoside resistance in *Enterobacteriaceae*, particularly, against amikacin [9,10,11]. 

The resistance mechanism developed against aminoglycosides is an enzymatic modification of the molecule by the aminoglycoside-modifying enzymes (AME), particularly, 6′-*N*-acetyltransferase type Ib [AAC(6′)-Ib], *O*-nucleotidyltransferases (ANT) and *O*-phosphotransferases (APH) [12,13]. Among the AME resistance genes, *Aac(6′)-Ib*, *Aac(3)-II*, *Ant(3″)-Ia*, and *Aph(3′)-Ia* are the most widely distributed ones [14].

Recently, a high rate of co-resistance involving the above-mentioned antimicrobials and β-lactams, particularly meropenem, has been shown in *Enterobacteriaceae* [15]. 

Carbapenemases are classified in three of the four classes of Ambler classification: class A carbapenemases use a serine residue to hydrolyze β-lactams, and they are encoded by *bla*_KPC_ gene; class B are zinc-dependent metallo-β-lactamases, and they are encoded by *bla*_VIM_, *bla*_IMP_, an *bla*_NDM_ genes; class D includes members of the OXA-encoding (*bla_OXA-48_*_-like_) genes [16,17]. Plasmid-mediated genes that are related to the three considered antimicrobial classes have been increasingly found to be in association [18].

For the first time since 2011, the antimicrobial usage in Europe from 2016 to 2018 was lower in food-producing animals than it was in humans [19]. Despite this encouraging data, antimicrobial resistance needs to be monitored in animals, and in particularly, in food-producing animals [20].

*Escherichia coli (E. coli)* are both pathogenic and commensal bacteria that are considered to be in food hygiene an important sign of fecal contamination. They are also used as sentinel bacteria to assess the presence of antimicrobial resistance phenomenon in animals and in human. *E. coli* can thus be isolated from a variety of sources, such as the feces, manure, water and foods of animal and plant origin, and consequently, they can easily survive in various environments and spread [21].

Meat and meat products can be contaminated in different stages of the food chain, from the abattoir during evisceration to the processing stage [22]. Consumers can come into contact with antimicrobial-resistant (AMR) *E. coli* during meat manipulation and preparation, thus leading to antimicrobial resistance spread. The cross-contaminations of the food environment, not well cooking meat and the increasing raw meat demand exacerbate the phenomenon [23]. In fact, many genetic similarities have been shown among bacteria to be found in food-producing animals and in humans, and particularly, in *E. coli* [24].

Antimicrobial surveillance programs (for example, the European Food Safety Authority program) include indicator microorganisms such as commensal *E. coli* because of their ubiquitous behavior which facilitates a proper trend analysis, assuming that this bacterium is intrinsically susceptible to the antimicrobial molecule that is being considered [25]. Furthermore, *E. coli* can transfer plasmid-mediated resistance to other bacteria that are both commensal and pathogenic, thus causing the occurrence of hard-to-treat infections [26].

Antimicrobial resistance is frequently detected in bacteria that are isolated from food producing animals because of their extensive and inappropriate use of antibiotics as growth promoters or as preventive treatments [27], thus resulting in the emergence of antimicrobial resistance in foodborne pathogens and commensal bacteria in the food of animal origin [28]. Pig production is an intensive industry for which the use of antibiotics for treatment of various diseases is widely diffused [27]. In addition, the current scenario of the anthropogenic transformation of the landscape involves forcing wildlife into greater contact with humans and their livestock, thereby increasing the risk of antimicrobial resistance transmission through different populations [29]. Antimicrobial resistance has already been reported in commensal bacteria in wildlife [30] and wild boar hunting, and the processing of their meat has been reported to be a possible route of the transmission of AMR bacteria and resistant genes to humans [23]. The dramatic increase in the contact between domestic and wild animal species would necessitate the monitoring of this phenomenon. In this context, the comparison of pork vs. meat from the wild is of interest.

Moreover, the food-producing environment plays an important role in the dissemination of AMR bacteria [2], and the production of biofilm, which are microbial communities that live together in a self-made extracellular polymeric matrix, may facilitate this phenomenon. The biofilm formation process establishes a gradient of available substances into the matrix, thereby creating an aerobic and metabolically active outer layers and an anaerobic and low-nutrient inner layer. Quinolones, β-lactams and aminoglycosides’ are not active to be in anaerobic conditions, thus causing a failure of the antimicrobial action. Moreover, they are able to kill rapidly dividing cells in the outer layer, thus causing the overwhelming of slow bacteria cell growth, leading to antimicrobial tolerance [31]. This physical barrier allows for the exchange of genetic elements among microorganisms [32], for example, both commensal and pathogenic bacteria [33].

The aim of the present study was to evaluate the prevalence of AMR *E. coli* that was isolated from pork and wild boar meat products and evaluate the involvement of food of animal origin in the spread of this phenomenon. The key goal of the study was to compare the antimicrobial resistance between the strains that were isolated from meat. In particular, the antimicrobial-treated farmed pigs and antimicrobial-free wild ones from around the world were compared. Ciprofloxacin and nalidixic acid (quinolones), amikacin (aminoglycosides) and meropenem (β-lactams) resistances were detected (phenotypic and genotypic analysis). Particular attention was paid to the plasmid-mediated resistance genes and their simultaneous presence in resistant strains. Moreover, the biofilm production was evaluated.

## 2. Materials and Methods

### 2.1. Sample Collection

Between January 2018 and January 2020, 1003 pork products (sausages, cotechino, salami, meatballs, meat skewers) from meat-processing companies and 1052 wild boar fresh meat samples from slaughterhouses were collected by the Istituto Zooprofilattico Sperimentale della Lombardia e Emilia Romagna (IZSLER), which is located in Reggio Emilia (44°42′34”56 N, 10°37′13”80 E).

### 2.2. Escherichia coli Isolation and Counting

According to the ISO standard 16649-2:2001 [34], *E. coli* was isolated from the meat samples. Briefly, 10 g of meat was homogenized 1:10 in Buffered Peptone Water (BPW), 1 mL was included in Triptone Bile-X-Gluc (TBX) agar medium, and it was incubated at 44 ± 1 °C for 18–24 h. The typical blue-green colonies were counted, selected, and subjected to an indole test. Positive colonies were confirmed as being *E. coli* using API 20E miniaturized system (bioMérieux, Marcy-l’Étoile, France). The methods have been accredited according to the ISO standard 17025:2018 [35]. The cut-off considered for the further analysis was over 10 CFU/g of *E. coli* colonies.

The *E. coli* isolates were then sent to the laboratory of Food Hygiene and Inspection of the Veterinary Science Department, University of Parma.

### 2.3. Antimicrobial Resistance

All of the isolated *E. coli* were analyzed through Kirby–Bauer test to evaluate their phenotypical AMR profile. A bacterial suspension of 1.5 × 10^8^ CFU/mL (0.5 McFarland) was used to seed the Mueller–Hinton Agar plates (Biolife Italiana, Milan, Italy) following the EUCAST guidelines [36]. An antibiotic disk (NeoSensitabs, Rosco Diagnostica, Denmark) of ciprofloxacin 5 µg (CIPR), nalidixic acid 30 µg (NAL), amikacin 30 µg (AMI) and meropenem 10 µg (MERO) was applied onto the seeded plates and incubated at 37 ± 1 °C for 18–24 h. The inhibition diameter was evaluated, and the isolates were classified as being “resistant” (inhibition zone CIPR ≤ 16 mm, NAL ≤ 20 mm, AMI ≤ 16 mm, MERO ≤ 14 mm) “intermediate” (inhibition zone 17 mm ≤ CIPR ≤ 19 mm, 24 mm ≤ NAL ≤ 21 mm, 17 mm ≤ AMI ≤ 19 mm, 15 mm ≤ MERO ≤ 17 mm) or “sensible” (inhibition zone CIPR ≥ 20 mm, NAL ≥ 25 mm, AMI ≥ 20 mm, MERO ≥ 18 mm) following the EUCAST guidelines [36] and the CLSI guidelines [37] for NAL.

### 2.4. Genotypic Analysis

The phenotypically resistant strains were tested using end-point PCR for the detection of resistant genes. Five colonies of overnight bacterial culture on Triptic Soy Agar (TSA-Oxoid, Basingstoke, United Kingdom) were suspended in 1 mL of sterile distilled water. The DNA extraction was performed by heating it at 95 °C for 10 min, and the cellular debris were removed by 15,000 rpm centrifugation for 5 min. The supernatant was used for amplification after its proper quantification using a Biospectrometer Basic Eppendorf (Eppendorf, Milan, Italy).

The ciprofloxacin and nalidixic acid-resistant *E. coli* were tested to evaluate the presence of the chromosomal genes mutations. Particularly, *gyrA Ser83 (gyrA83), gyrA Asp87 (gyrA87)* and *parC Ser80 (parC80)*, *parC Glu84 (parC84)* were detected by MAS-PCR following Onseedaeng and Ratthawongjirakul’s [38] protocol with modifications that are described in Table 1.

The presence of PMQR (*qnrA, qnrB and qnrS*) was detected firstly in the quinolone-resistant strains and in *E. coli* harbouring at least one of the amikacin or meropenem plasmid-related genes to evaluate their co-presence. The multiplex PCR was set up following Salah et al.’s [39] protocol with variations (Table 1).

The amikacin plasmid resistance genes (*Aac(6′)-Ib*, *Aac(3)-II*, *Ant(3″)-Ia*, and *Aph(3′)-Ia)* were detected in the resistant *E. coli* by Shi et al.’s [14] multiplex PCR protocol with modifications including the *Aac(6′)-Ib* forward primer design (BLAST, NCBI) (Table 1). To evaluate their co-presence, the AMEs genes were detected also in the isolates that harbored PMQR or meropenem plasmid-resistant genes.

The PCR products were evaluated by electrophoresis with 2% of agarose with SYBR Safe DNA gel stain (Invitrogen, Poland, OR).

Following the previous criterion, the multiplex PCR for the detection of *bla*_KPC_, *bla*_VIM_, *bla*_IMP_, *bla*_NDM_ and *bla_OXA-48_*_-like_ was performed using Doyle et al.’s [40] protocol with some variations (Table 1).

The PCR products were evaluated by electrophoresis with 1.5% of agarose with SYBR Safe DNA gel stain (Invitrogen, Poland, OR).

All of the amplicons were visualized by UV light. A 100-bp DNA ladder from Promega s.r.l., Milan, Italy was used as a marker. Positive, negative and no template controls were included.

### 2.5. Biofilm Evaluation

The resistant *E. coli* were tested for the ability to produce a biofilm. The protocol for the biofilm formation on a 96-well plate was developed following the one by O’Toole [41]. The colonies were first suspended in BPW that was added with 1% glucose and incubated at 37 ± 1 °C for 24 h. An aliquot was taken and regenerated in a new BPW with 1% glucose until there was a concentration of 1.5 × 10^8^ CFU/mL (0.5 OD). Two hundred µL of bacterial suspension was placed in a 96-well plate. Each sample test was replicated 15 times. The positive control was *E. coli* ATCC 25922, which is a strong biofilm producer, and the blank was set up as culture broth, only. After incubation at 37 ± 1 °C for 24 h, the plates were washed by immersion in distilled water, dried upside down at 42 ± 1 °C for 20 min, and finally, they were colored with 100 µL of crystal violet for 5 min. The plates were washed three times by immersion in distilled water, and they were allowed to dry at 42 ± 1 °C for 1 h. The colonies were resuspended with 130 µL of ethanol per well, and the data were read on a spectrophotometer (Multiskan FC Version 1.00.75, Thermofisher Scientific, Waltham, MA, USA) at a wavelength of 620 nm.

The data were used to classify bacteria into 4 categories:-Non-adherent ones if the optical density (OD) was lower than the optical density of the blank wells (ODc);-Weak biofilm producers if ODc ≤ OD < 2XODc;-Moderate biofilm producers if 2XODc ≤ OD < 4XODc;-Strong biofilm producers if OD ≥ 4XODc.

### 2.6. Statistical Analysis

To evaluate the statistical difference between the variables that were considered, the *p* value was calculated (MedCalc Software Ltd.–free version, Ostend, Belgium). Particularly, the antimicrobial resistance prevalence that was obtained from the analysis of the pork and wild boar *E. coli* was compared using a Chi-Square test, and a *p* value < 0.05 was considered to be statistically significant.

The data collected must respect the following relation to be statistically considered:n > 30; np > 5, n (1 − p) > 5n = the number of animals;p = the proportion of *E. coli* strains with the characteristics that are being studied.

## 3. Results

The *E. coli* were isolated from pork meat and wild boar meat products. Following the protocol that is described above, 198 *E. coli* were selected from the count plates which were obtained from 1003 processed pork meat samples (19.6%; CI 95% = 18.2–21.2), and 221 strains were selected from 1052 processed wild boar meat (21%; CI 95% = 18.5–23.5). When we were comparing the number of isolates between the two groups, no statistical difference was found (*p* = 0.4).

### 3.1. AMR E. coli Phenotypic Profile

The *E. coli* isolates were tested for their antimicrobial resistance against quinolones (ciprofloxacin and nalidixic acid), aminoglycosides (amikacin) and β-lactams (meropenem).

In the pork, the resistant *E. coli* strains to nalidixic acid and ciprofloxacin showed an occurrence rate of 41.9% (83/198; CI 95% = 37.9–45.9) and of 5.5% (11/198; CI 95% = 2.3–8.7), respectively.

The quinolone resistance prevalence in the *E. coli* that was isolated from wild boars followed a different trend: it was 36.2% (80/221; CI 95% = 34.3–38) for nalidixic acid and 0.4% (1/221; CI 95% = 0–3) for ciprofloxacin.

The resistance to ciprofloxacin was matched in all of the cases with the resistance to nalidixic acid. The difference between the prevalence of the quinolone-resistant *E. coli* in the pork and in the wild boars was statistically significant only for ciprofloxacin resistance (*p* = 0.0046).

A resistance to amikacin (aminoglycosides) was detected in 6.6% (13/198; CI 95% = 3.3–9.9) of the *E. coli* that were isolated from pork and in 1.8% (4/221; CI 95% = 0–5.38) of the wild boar meat isolates. The difference was statistically significant (*p* = 0.044). No resistant profiles were detected against meropenem (β-lactam).

Five *E. coli* strains from pork and two from wild boar meat were resistant both to quinolones and amikacin.

The intermediate resistant profiles were also evaluated. The prevalence of quinolone intermediate profiles was 54.5% (108/198; CI 95% = 47.6–61.4) and 62.9% (139/221; CI 95% = 56.63–69.4) for *E.coli* isolated from the pork and wild boar samples, respectively. Particularly, the pork isolates showed intermediate resistance only to nalidixic acid. The wild boar strains mainly showed intermediate profiles against nalidixic acid, but seven strains had simultaneous intermediate profiles to both of the quinolones that were tested.

The intermediate AMR profile to amikacin was observed in 38.9% of the pork *E. coli* strains (77/198; CI 95% = 34.9–42.9) and in 44.3% of the wild boar strains (98/221; CI 95% = 42–46.6).

The AMR intermediate profile of both of the antimicrobial classes was found in 38 pork strains and 54 wild boar strains. No intermediate profile was detected for meropenem. A resistance to nalidixic acid was frequently associated with an intermediate amikacin profile (13.6% pork *E. coli* and 19% in wild boar *E. coli*). All of the AMR patterns, both for the resistant and intermediate profiles, are shown in Table 2.

### 3.2. AMR E. coli Genotypic Profile

The strains showing phenotypic resistance to quinolones were analyzed to evaluate the presence of both the chromosomal and plasmid resistance genes. The chromosomal point mutations were firstly evaluated: in the pork samples, the quinolone-resistant *E. coli* showed a gyrA83 mutation in 13.1% of them (11/84), gyrA87 in 44% of them (37/84), parC80 in 7.1% of them (6/84), and parC84 in 1.2% of them (1/84). In the wild boar samples, the quinolone-resistant *E. coli* in 2.5% of the isolates (2/80) harbored only the gyrA87 mutation.

The plasmid-mediated quinolone resistance gene *qnrA* was detected in 1.2% (1/84) of the pork strains and never from the wild boars ones, while *qnrB* was present in 2.4% (2/84) of the pork strains and 8.7% (7/80) of the wild boar isolates. *QnrS* was found in 13.1% (11/84) of the pork *E. coli* strains and the 7.5% (6/80) of the wild boar isolates.

The strains showing a phenotypic resistance to amikacin were tested for the presence of the most common plasmid genes: 38.5% (5/13) of the pork *E. coli* harbored *Ant(3″)-Ia* and the 30.8% of it harbored (4/13) *Aac(6′)-Ib.* One of the strains simultaneously harbored the *Aac(3)-II* and *Aph(3′)-Ia* genes. All of the wild boar *E. coli* which were phenotypically resistant to amikacin were genotypically confirmed as harboring the *Aac(6′)-Ib* gene. Independently from their phenotypical expression, the strains that harbored PMQR were tested for the presence of amikacin-resistant genes and vice versa. In the pork samples, the co-presence of AMEs and PMQR genes was observed in 13 strains with different phenotypical AMR profiles: four strains were resistant to quinolone and amikacin, and nine strains were resistant to quinolone, but they were susceptible to amikacin. Table 3 shows all of the gene patterns that were found. Twelve wild boar *E. coli,* which were resistant to quinolones with PMQR genes, co-harbored AME genes with a sensible phenotypic profile of amikacin (Table 3).

### 3.3. Resistant E. coli and Biofilm Production

The resistant *E. coli* were tested for the ability to produce a biofilm. In the pork samples, 46/91 (50.6%; CI 95% = 40.3–60.8) of them were biofilm-producing *E. coli*. In particular, 29/91 (31.9%; CI 95% = 26–37.8) of them were weak, 12/91 (13.2%; CI 95% = 10.3–16) of them were moderate and 5/91 (5.5%; CI 95% = 0.8–10.2) of them were strong biofilm producers. Among the strains that were isolated from the wild boar meat, 52 out of 81 of them (64.1%; CI 95% = 61.1–67.2) were able to produce a biofilm. In particular, 16/81 of them (19.7%; CI 95% = 14.1–25.3), 24/81 of them (29.6%; CI 95% = 19.7–39.5) and 12/81 of them (14.8%; CI 95% = 7.1–22.5) were weak, moderate and strong biofilm producers, respectively. No statistically significant difference was found between the two groups (pork *E. coli* and wild boar *E. coli*) that were tested (*p* = 0.07).

In 29/46 (63%; CI 95% =49–76.95) of the pork *E. coli* which were able to produce a biofilm, antimicrobial resistance genes were found. The prevalence of this in wild boar *E. coli* was 7/29 (24.1% CI 95% =8.5–39.7), with a *p* value of 0.0145.

## 4. Discussion

Meat consumption is acknowledged to be a possible route of antimicrobial resistance transmission to humans, and food-producing animals are recognized as an important reservoir of various resistant bacteria [40,42]. Among them, *E. coli* is used as sentinel in surveillance programs due to their ability to acquire antimicrobial resistance. Multinational surveillance is essential for the early detection of increasing resistance patterns across countries [43,44].

This study focused on the pork production chain with the aim to compare the antimicrobial resistance prevalence in *E. coli* that had been isolated from both pork meat and wild boar meat and to evaluate the effect of selective pressure in domestic vs. wild animals.

The data showed that the prevalence of AMR *E. coli* isolates was higher in the pork meat (45.9%) than it was in the wild boar meat (37.1%). In particular, the resistance prevalence against ciprofloxacin, nalidixic acid and amikacin was, respectively 5.1%, 5.7% and 4.8% higher in the pork isolates. For ciprofloxacin and amikacin resistance, the difference was statistically significant. Despite this, the percentages of antimicrobial-resistant bacteria are high in both groups, and this may be due to indirect contact between the two habitats such as the use of pig manure in agriculture and the consequent dispersion of AMR bacteria [45].

Being different from the results obtained in this study, an EFSA/ECDC report [46] showed lower levels of resistance to these antimicrobials in pigs, even though there was a wide variation among the reported countries for both ciprofloxacin and nalidixic acid resistance. No *E. coli* which were resistant to meropenem were found in our study, and the same result has been confirmed throughout Europe, as reported by the EFSA/ECDC report [46].

In the present study, the intermediate resistant profiles were higher in the wild boar *E. coli* (82.8%) vs. the pork *E. coli* (74.2%), suggesting that the lower selective pressure on wildlife can affect the antimicrobial resistance expression in the bacteria [23]. This is also confirmed by the ciprofloxacin intermediate profile which was shown only in the wild boar isolates. In addition, the intermediate resistance values to amikacin and to nalidixic acid in the wild boar meat isolates were higher than they were in the ones (differences of 5.4% and 8.4%, respectively).

All of the resistant *E. coli* strains were genotypically tested in order to evaluate only the most common chromosomal mutations and plasmid-mediated resistance genes.

The quinolone-resistant genes were found in 58.3% of the phenotypically resistant pork *E. coli*. All of the tested genes, both the chromosomal mutations and PMQRs, have been found in the considered samples, and the most frequent mutation was *gyrA87*, which was followed by *gyrA83* and *qnrS*. These data are similar to those that were found in other European countries [47,48,49].

In the wild boar meat isolates, only *gyrA87, qnrB* and *qnrS* were found in the AMR strains (17.5%), and despite the data that have been reported on regarding pork *E. coli*, *qnrB* was the gene that was most frequently present. These findings have been reported in another study on wildlife species, even though their meat was not considered by the authors [50]. In the present study, the confirmation rates of the genotypic resistances were lower than the phenotypic resistances were. This finding suggests the possible presence of other chromosomal genes mutations, plasmid-mediated gene variants or aspecific mechanisms causing quinolone AMR [7].

The percentages of AMR *E. coli* were genotypically confirmed to be higher for amikacin resistance than they were for a resistance to quinolones; 76.9% of the pork *E. coli* harbored AMEs genes, especially *Ant(3″)-Ia* and *Aac(6′)-Ib*, which was reported also by Poirel et al. [49]. All of the wild boar amikacin-resistant *E. coli* harbored AMEs gene, particularly *Aac(6′)-Ib*. Additionally, Poirel et al. [49] and Mercato et al. [51] reported that the *Aac(6′)-Ib* gene was the most frequently found one in game animals.

Plasmid-mediated resistance genes were taken into account in this study due to their association to bacteria, as reported by EFSA [18]: quinolones PMQR, AME genes and Carbapenemases resistance genes seem to be frequently co-present and horizontally transferred from one bacterium to another simultaneously [18]. In this study, the phenotypically resistant *E. coli* that harboured at least one of the plasmid-mediated resistance genes, which were related to the three antimicrobial considered classes, were tested for the other plasmid genes. The data demonstrated that the gene was co-present only for quinolones and aminoglycosides, and on the contrary, no carbapenems resistance genes were found. Moreover, the prevalence of co-present plasmid genes was similar between the pork and wild boar *E. coli* (13 and 12 strains, respectively), and the most frequent co-present gene patterns were *qnrS* and *Ant(3″)-Ia*. These data differ from other findings which demonstrated a stronger correlation between *qnrA* and *qnrB* with multidrug resistance in the enterobacteria [52]. *QnrS* was found to be frequently associated with *Aac(6′)-Ib* [53]. Otherwise, Rodriguez-Martinez et al. [52] confirmed the difficult association between the co-presence of *qnr* and carbapenems-related genes.

The majority of wild boar resistant *E. coli* isolates (64.2%) were able to produce a biofilm, whilst approximately half of the pork isolates (50.6%) demonstrated this ability. Moreover, the wild boar isolates were frequently moderate biofilm producers, while the pork *E. coli* were mostly weak biofilm producers. The rate of biofilm production has been reported as being high (60%) in bacteria that has been isolated from other wild animals, even if the available data frequently depend on the strain origin, cell membrane, surfaces, culture medium and methodology that were used [54]. Thanks to the biofilm matrix, biofilm-producing bacteria can tolerate harsh living conditions in the environment [55], and at the same time, this situation can promote the exchange of genetic elements, thus facilitating the spread of antimicrobial resistance through the bacteria [56]. In fact, different studies have demonstrated that the majority of bacteria which are able to produce biofilm are MDR [54,56]. The data that were generated in this study showed the high ability of biofilm formation in resistant bacteria, both in pork and wildboar meat. Moreover, the antimicrobial resistance genes were found to be in high percentages in pork, but the association of the biofilm production ability is statistically significant only in wild boar *E. coli*. The difficult management of wild boar carcasses in game-handling establishments [57], together with the ability of the microorganisms to produce a biofilm, could be a possible factor causing environmental cross-contamination and the spreading of AMR. To the authors’ knowledge, the present study is the first to report the presence of AMR *E. coli* that were isolated from fresh wild boar meat and conduct an evaluation of their biofilm production ability.

## 5. Conclusions

The study evaluated the prevalence of AMR strains among fresh meat samples. The role that is played by the farmed vs. wild living conditions of food-producing animals was evaluated. Contrary to our expectations, many isolates from the antimicrobial-free world showed to have high antimicrobial resistance percentages as well as the strains from domestic animal meat products. The differences in the genotypical AMR confirmations were highlighted between the two animal species that were tested. This evidence suggested that antimicrobial resistance spreads, and its development can be strongly influenced by the environmental selective pressure that is on animals, and consequently, on their derived food products. Meat processing can reduce the hazard of AMR bacteria, but at the same time, cross-contaminations in the food environment may still represent a risk of transmission to consumers. In addition, the ability to form a biofilm could raise the possibility of AMR transmission through food consumption. Food safety and quality assurance that are related to spread of antimicrobial resistance can be at risk, as confirmed by our data.

Finally, humans can both positively or negatively affect this phenomenon by controlling the antimicrobials usage and by following good manufacturing and hygienic practices during food production Despite this, uncontrolled risk factors can occur affecting both domestic and wild food production chains.

## Figures and Tables

**Table 1 foods-11-03662-t001:** Oligonucleotide primers and PCR conditions used in this study.

Genes	Sequences	Size bp	PCR Conditions		Reference
Quinolone Resistance-Determining Regions					
** *gyrA* **	**F** 5′-TACACCGGTCAACATTGAGG-3′ **R** 5′-TTAATGATTGCCGCCGTCGG-3′	647	Denaturation 94 °C for 5 min, 30 cycles 94 °C for 30 s, 60 °C for 1 min, 74 °C for 2 min, and a final extension at 74 °C for 10 min	Final volume 25 μL: 2x Green GoTaq Flexi Buffer, 1.5 mM of MgCl_2_, 0.2 mM of dNTPs, 1 U of GoTaq G2 Flexi DNA Polymerase, primers at 1 μM,1 μL of sample lysate, Nuclease Free Water to final volume.	[38]
** *gyrA 83* **	**F** 5′-TACCATCCCCATGGTGACTC-3′	440			
** *gyrA 87* **	**R** 5′GCCATGCGGACAATCGTGTC-3′	255			
** *parC* **	**F** 5′-AAACCTGTTCAGCGCCGCATT-3′ **R** 5′-GTGGTGCCGTTAAGCAAA-3′	395		Final volume 25 μL: 2x Green GoTaq Flexi Buffer, 4 mM of MgCl_2_, 0.4 mM of dNTPs, 1.25 U of GoTaq G2 Flexi DNA Polymerase, primers at 0.4 μM, 1 μL of sample lysate, Nuclease Free Water to final volume.	
** *parC 80* **	**F** 5′-AATACCATCCGCACGGCGATAG-3′	289			
** *parC 84* **	**R** 5′CGCCATCAGGACCATCGGTT-3′	153			
** *uspA* **	**F** 5′- CCGATACGCTGCCAATCAGT-3′ **R** 5′-ACGCAGACCGTAGGCCAGAT -3′	884		uspA was added as internal control at the conditions described above	
**Plasmid-Mediated Quinolone Resistance Genes**					
** *qnrA* **	F: 5′-ATTTCTCACGCCAGGATTTG-3′ R: 5′-GATCGGCAAAGGTTAGGTCA-3′	515	Denaturation 92 °C for 5 min, 30 cycles 95 °C for 45 s, 58 °C for 45 s, 72 °C for 1 min, a final extension at 72 °C for 10 min	Final volume 50 μL: 1x Green GoTaq Flexi Buffer, 2.5 mM of MgCl_2_, 0.2 mM of dNTPs, 1.5 U of GoTaq G2 Flexi DNA Polymerase, primers at 0.5 μM,1 μL of sample lysate, Nuclease Free Water to final volume.	[39]
** *qnrB* **	F: 5′-GATCGTGAAAGCCAGAAAGG-3′ R: 5′-ACGATGCCTGGTAGTTGTCC-3′	469			
** *qnrS* **	F: 5′-ACGACATTCGTCAACTGCAA-3′ R: 5′-TAAATTGGCACCCTGTAGGC-3′	417			
**β-lactams resistance genes**					
** *bla* _KPC_ **	F: 5′- TGTCACTGTATCGCCGTC-3′ R: 5′-CTCAGTGCTCTACAGAAAACC-3′	900	Denaturation 95 °C for 5 min, 35 cycles 95 °C for 45 s, 62 °C for 30 s, 72 °C for 1 min, and a final extension at 72 °C for 8 min	Final volume 50 μL: 1x Green GoTaq Flexi Buffer, 2 mM of MgCl_2_, 0.2 mM of dNTPs, 2 U of GoTaq G2 Flexi DNA Polymerase, primers *bla*_KPC_, *bla*_IMP_, *bla*_VIM_ at 0.3 μM, primers *bla*_NDM_ at 0.4 μM, primers *bla*_oxa-48-like_ 0.5 μM, 1 μL of sample lysate, Nuclease Free Water to final volume.	[40]
** *bla_I_* _MP_ **	F: 5′-GAAGGCGTTTATGTTCATAC R: 5′-GTACGTTTCAAGAGTGATGC-3′	587			
** *bla* _VIM_ **	F: 5′-GTTTGGTCGCATATCGCAAC-3′ R: 5′-AATGCGCAGCACCAGGATAG-3′	389			
** *bla* _NDM_ **	F: 5′-GCAGCTTGTCGGCCATGCGGGC-3′ R: 5′-GTCGCGAAGCTGAGCACCGCAT-3′	782			
** *bla* _oxa-48-like_ **	F:5′-GCGTGGTTAAGGATGAACAC-3′ R:5′-CATCAAGTTCAACCCAACCG-3′	438			
**Aminoglycosides modifying enzyme-encoding genes**					
** *Aac(6′)-Ib* **	**F** 5′-CCCAGTCGTACGTTGCTCTT-3′ R 5′- AAACCCCGCTTTCTCGTAGC-3′	239	Denaturation 95 °C for 5 min, 30 cycles 95 °C for 30 s, 63 °C for 1 min, 72 °C for 2 min, and a final extension at 72 °C for 10 min	Final volume 50 μL: 1x Green GoTaq Flexi Buffer, 3 mM of MgCl_2_, 0.2 mM of dNTPs, 2 U of GoTaq G2 Flexi DNA Polymerase, primers at 0.4 μM except for *Ant(3”)-Ia* at 0.3 μM, 1 μL of sample lysate, Nuclease Free Water to final volume.	[14]
** *Ant(3″)-Ia* **	**F** 5′-CCGGTTCCTGAACAGGATC-3′ R 5′-CCCAGTCGGCAGCGACATC-3′	180			
** *Aph(3′)-Ia* **	**F** 5′-CAAGATGGATTGCACGCAGG-3′ **R** 5′-TTCAGTGACAACGTCGAGCA-3′	317			
** *Aac(3)-II* **	**F** 5′-GCTCGGTTGGATGACAAAGC-3′ **R** 5′-AGGCGACTTCACCGTTTCTT-3′	379			

**Table 2 foods-11-03662-t002:** Phenotypic resistant and intermediate profiles of *E. coli* isolated from pork and wild boars meat sample.

No Pork *E. coli* (%)			Quinolones							Total
			Resistant Profile			Intermediate Profile			None	
			CIPRO	NAL	CIPRO + NAL	CIPRO	NAL	CIPRO + NAL		
**Aminoglycosides**	**Resistant profile**	**AMI**	-	2 (1%)	3 (1.5%)	-	7 (3.5%)	-	1 (0.5%)	13 (6.6%)
	**Intermediate profile**	**AMI**	-	27 (13.6%)	6 (3%)	-	38 (19.1%)	-	6 (3%)	77 (38.9%)
	**None**		-	43 (21.7%)	2 (1%)	-	63 (31.8%)	-	-	108 (54.5%)
	**TOTAL**		-	72 (36.4%)	11 (5.5%)	-	108 (54.5%)	-	7 (3.5%)	**198**
**No Wild boars *E. coli* (%)**										
**Aminoglycosides**	**Resistant profile**	**AMI**	-	1 (0.4%)	1 (0.4%)	-	2 (0.9%)	-	-	4 (1.8%)
	**Intermediate profile**	**AMI**	-	42 (19%)	-	-	50 (22.6%)	4 (1.8%)	2 (0.9%)	98 (44.3%)
	**None**		-	36 (16.3%)	-	-	80 (36.2%)	3 (1.4%)	-	119 (53.8%)
	**TOTAL**		-	79 (35.7%)	1 (0.4%)	-	132 (59.7%)	7 (3.2%)	2 (0.9%)	**221**

**Table 3 foods-11-03662-t003:** Plasmid-mediated genes co-presence in E. coli isolated from pork and wild boar meat.

No Pork *E. coli*	Quinolones			Aminoglycosides			
	*qnrA*	*qnrB*	*qnrS*	*Aac(3)-II*	*Aac(6′)-Ib*	*Ant(3”)-Ia*	*Aph(”)-Ia*
1			x				
3					x		
2						x	
1		x			x		
1		x				x	
3			x		x		
6			x			x	
1					x	x	
1	x				x	x	
1			x	x			x
No Wild boar *E. coli*							
1		x					
3					x		
2		x				x	
5			x			x	
1					x	x	
4		x			x	x	
1			x		x	x	

## Data Availability

Data are reported within the article.

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
