# Peer review of "Antimicrobial Resistant E. coli in Pork and Wild Boar Meat: A Risk to Consumers"

_foods, 2022, doi:10.3390/foods11223662_

Round 1
Reviewer 1 Report
Dear Editor and authors,
The manuscript (Antimicrobial resistant E. coli in pork and wild boar meat: a risk to consumers) needs many corrects and modifications.
1-The manuscript abstract includes many scientific names of bacteria. Scientific names should be written in italics throughout the manuscript.
2-The introduction needs to add paragraphs about the presence of E. coli bacteria in meat and meat products, see (1-Rega, M., Carmosino, I., Bonilauri, P., Frascolla, V., Vismarra, A., & Bacci, C. (2021). Prevalence of ESβL, AmpC and Colistin-Resistant E. coli in meat: a comparison between pork and wild boar. Microorganisms, 9(2), 214.
2-Al-kuzayi, A. K., & Al-Sahlany, S. T. (2011). DETECTING FOR E. COLI O157: H7 IN DAIRY PRODUCTS WHICH WERE LOCALLY PROCESSED AND FOUND IN BASRA CITY MARKETS. Basrah Journal of Agricultural Sciences, 24(1).
3-The purpose of the manuscript needs to be modified and clarified.
4- Page 3 line 115, correct the bacterial numbers unit to CFU/ mL, This method used broth media.
5-Antimicrobial resistance method is unclear, How many the bacteria number cultured in culture media??
6-Tables 1 and 2 can be horizontal rather than vertical, which is better for a manuscript.
7-The conclusions of the current study should be separated from the discussion and placed in a special chapter under the discussion chapter.
Reviewer 2 Report
This paper discusses the prevalence of antimicrobial resistant E. coli isolated from pork and wild boar meat products in a 2018-2020 period. In my opinion, the main contribution of the paper is a fact that many isolates from an antimicrobial-free world (wild animals) showed high antimicrobial resistance. In general, paper is well written although I would like to see some numerical data (minimum inhibitory concentration) for at least some of the samples, given that Kirby Bauer methodology offers rough estimation of AMR. Also, I feel that section describing biofilm synthesis and pertaining discussion should be explained better, or in some more detailed manner.
Reviewer 3 Report
Dear Authors,
The aim of the presented study was to evaluate the prevalence of AMR E. coli isolated from pork and wild boar meat products and to compare the spread of this phenomenon between animals regularly treated with antimicrobials and the antimicrobial -free world of wildlife.
The introduction of the presented manuscript provides a good, generalized background of the topic that quickly gives the reader an appreciation of the scientific relevance and timeliness of the research theme. The motivations for this study are very clear. The manuscript was written really thoroughly, when it comes to both, description of research material. However, the used statistical methods should be revised and completed with other more advanced tools.
I have some suggestions for improvement for Authors which are as follows:
ü In the whole manuscript check the word “E. coli” (for instance: line 257). It should be written in italic.
ü Lines 109; 114: please add a word “ISO standard”
ü In the subsection no. 2.6., please complete information about used statistical program (name, producer, country).
ü Indicate the importance of the obtained results in terms of food quality assurance and management systems.
ü Please recheck thoroughly the whole article and improve its grammatical mistakes.
ü Please recheck references according to the journal guidelines.
Round 2
Reviewer 3 Report
Dear Authors,
The Authors have provided satisfactory responses to the reviewers' comments except for one point. Please complete the section References with standards:
- ISO standard 16649-2:2001 (cited in line 122,
- ISO standard 17025:2018 (cited in line 127).
Author Response
The section References has been completed with the cited standards.